# Insights into Abiotically-Generated Amino Acid Enantiomeric Excesses Found in Meteorites

**DOI:** 10.3390/life8020014

**Published:** 2018-05-12

**Authors:** Aaron S. Burton, Eve L. Berger

**Affiliations:** 1Astromaterials Research and Exploration Science, NASA Johnson Space Center, Houston, TX 77058, USA; 2GeoControl Systems, Jacobs JETS contract, NASA Johnson Space Center, Houston, TX 77058, USA; eve.l.berger@nasa.gov

**Keywords:** amino acids, chirality, meteorites, homochirality, enantiomer, prebiotic chemistry

## Abstract

Biology exhibits homochirality, in that only one of two possible molecular configurations (called enantiomers) is used in both proteins and nucleic acids. The origin of this phenomenon is currently unknown, as nearly all known abiotic mechanisms for generating these compounds result in equal (racemic) mixtures of both enantiomers. However, analyses of primitive meteorites have revealed that a number of amino acids of extraterrestrial origin are present in enantiomeric excess, suggesting that there was an abiotic route to synthesize amino acids in a non-racemic manner. Here we review the amino acid contents of a range of meteorites, describe mechanisms for amino acid formation and their potential to produce amino acid enantiomeric excesses, and identify processes that could have amplified enantiomeric excesses.

## 1. Introduction

Amino acids are the building blocks of proteins, the molecular machines responsible for catalyzing nearly all chemical reactions necessary for life. The universal protein alphabet consists of 20 amino acids, all of which are α-amino acids, where the amino group is attached to the carbon immediately adjacent to the carboxylic acid moiety (Figure 1A). Of these 20 proteinogenic amino acids, one is achiral (glycine), and the other 19 have chiral carbons and can exist as either of two possible stereoisomers (Figure 1B), which are denoted as enantiomers. Enantiomers of a given chiral compound have identical chemical and physical properties, except for how they interact with other chiral compounds and how they interact with polarized light. Nevertheless, biology has evolved to use only l-amino acids (homochirality) in the production of genetically encoded proteins. Similarly, biology’s informational polymers, ribonucleic and deoxyribonucleic acid, are also homochiral, containing only d-enantiomers (Figure 1C).

Given the physical and chemical equivalence of d- and l-amino acids (and sugars), there does not appear to be an a priori reason for the selection of l-amino acids over d-amino acids. Indeed, it has been shown that stereoisomers of the protein HIV protease 1 containing only d-amino acids or only l-amino acids exhibit identical catalytic properties, with the sole difference in their activities being an inversion in the stereochemistry of the substrates on which these two enzymes act [1]. Although most l-proteins have evolved to recognize d-sugars, studies have shown that naturally-occurring l-proteins can also act on l-sugars, including the catabolism of l-glucose [2], and l-gulose and l-fructose [3]. Thus, there does not appear to be a strict link between amino acid and sugar chirality. This is perhaps not surprising given that in the L and D terminology only one chiral center is evaluated to assign L or D, whereas the predominant sugars in biology contain multiple chiral centers (e.g., ribose, deoxyribose, glucose, fructose, etc.; Figure 1C). These observations, taken together, suggest that any of the four chiral combinations (l-amino acids and l-sugars; l-amino acids and d-sugars; d-amino acids and d-sugars; and d-amino acids and l-sugars) could have sustained life. The ubiquity of l-amino acids in proteins and d-sugars in nucleic acids and metabolism across all domains of life provides strong evidence that these choices were fixed prior to the Last Universal Common Ancestor (LUCA). In contrast with amino acids and sugars, both enantiomers of chiral phospholipids (a major component of cell membranes), are used in extant biology, with archaea and bacteria using opposite enantiomers [4]. In this case, selection occurred post-LUCA. While contemporary biology tells us which stereochemistries were ultimately selected for, and provides a constraint on when that selection occurred (prior to LUCA, >3.5 Ga for amino acids and sugars, and post-LUCA for phospholipids), analysis of modern biology reveals little about how and why l-amino acids and d-sugars were ultimately selected over their respective enantiomers, necessitating an alternative approach.

Direct evidence of prebiotic chiral selection on Earth has not yet been found. It is likely that any such records on Earth have been overwritten by billions of years of geological or biological processing. However, prebiotic chemistry studies in the lab have revealed the facile nature of amino acid synthesis under a broad range of plausibly prebiotic conditions. These studies include the spark discharge experiments pioneered by Miller and Urey [5,6], reductive aminations [7], aqueous Strecker-type chemistry [8], and Fischer-Tropsch type syntheses [9,10], etc. Chiral amino acids formed by these processes, however, are formed in equal (racemic) mixtures of l- and d-enantiomers. Hence, although these reactions could have provided a steady supply of amino acids for the origins of life, they do not appear to be capable of generating chiral excesses of any magnitude, let alone homochirality. Key outstanding questions in the origins of life, then, include what led to the transition from racemic, abiotic chemistry to the homochirality observed in biology, and whether this transition was a biological invention or was initiated by abiotic processes.

## 2. Meteorites and Prebiotic Chemistry

Although samples of Earth’s nascent prebiotic chemistry are absent, meteorites, the remnants of asteroids, comets and planetary bodies that reach the surface of Earth, preserve records of processes and conditions during the origin and early evolution of the Solar System. Meteorites in collections on Earth have come from an incredible range of parent bodies, spanning primitive asteroids to planet-sized objects including the Moon and Mars. Carbonaceous chondrites in particular have revealed that complex chemistry relevant to the origins of life was occurring on a range of planetary bodies in early solar system. Compound classes found in carbonaceous chondrites that are of particular interest for the origins of life include carboxylic acids [11], amino acids [12], sugars and sugar derivatives [13], and nucleobases [14]. Even among the carbonaceous chondrites, however, significant variations in the abundances and diversities of compounds have been observed, revealing an interplay between the inorganic meteorite parent body material, the pool of available reactants (e.g., ammonia, hydrogen cyanide, aldehydes and ketones), and secondary geological processes such as aqueous alteration and thermal alteration [15,16,17].

The carbonaceous chondrites have been subdivided into eight distinct groups (CI, CM, CR, CH, CB, CO, CV and CK) that are named after a representative meteorite (e.g., CI denotes Ivuna-like). Carbonaceous chondrites are composed of mixtures of fine-grained silicate mineral matrix material, in which additional components are embedded (Table 1). These include: solid iron and iron/nickel metal; sulfide minerals; refractory phases such as calcium aluminum inclusions (CAIs) and amoeboid olivine aggregates (AOAs); and chondrules, among the first bits of matter to condense into the solid phase in our solar nebula. The presence of chondrules distinguishes carbonaceous and other chondrites from less primitive partially or completely differentiated meteorites. Carbonaceous chondrite groupings are based on factors such as bulk rock composition (e.g., enrichments or depletions of elements relative to the solar photosphere; ratios of reduced to oxidized iron; ratios of refractory and volatile elements, such as Al/Zn vs. Al/Mn), texture, mineralogy, and mineral compositions. Oxygen isotope compositions are also used to classify meteorites, although this is less diagnostic for carbonaceous chondrites, as oxygen isotope compositions of some carbonaceous chondrite groups overlap. Secondary processes such as aqueous alteration and thermal metamorphism are captured by petrologic sub-typing. Carbonaceous chondrites are assigned a number from 1–6, where a type 3.0 meteorite experienced minimal parent body processing after accretion, types 1 to <3 experienced increasing aqueous alteration, and types >3 to 6 experienced increasing thermal alteration [18,19,20,21,22,23,24,25,26]. For all carbonaceous chondrite groups, only a subset of the total petrologic types is observed (e.g., for CR chondrites, samples found to date are of types 1–3; for CK chondrites known samples are of types 3–6), indicating the parent bodies of these meteorites predominantly experienced either aqueous alteration or thermal alteration. Water is a key solvent for many reactions of prebiotic interest, but initial (post-accretion) water contents of meteorites have proven challenging to tightly constrain. However, the general consensus is that most of the carbonaceous chondrite groups had water-to-rock ratios less than 0.5:1 [27,28], and that aqueous alteration in most meteorites occurred at temperatures between 0 °C and 150 °C.

A sufficient number of carbonaceous chondrites of each group and petrologic sub-type have now been analyzed for amino acids for the following conclusions to be safely drawn: (1) carbonaceous chondrites of the same group and petrologic type tend to have similar abundances and distributions of amino acids; (2) within carbonaceous chondrite groups, increases in the degree of aqueous alteration in the meteorite parent body tend to reduce the total abundances of amino acids, and change the amino acid distribution; (3) carbonaceous chondrites from aqueously altered groups contain higher abundances of amino acids than thermally altered carbonaceous chondrite groups; and (4) thermally-altered meteorites have lower abundances of amino acids and a strong tendency for straight-chain amino acids where the amino group is on the carbon farthest from the carboxylic acids (*n*-ω-amino acids; see Figure 2) [16,17].

Because amino acids are ubiquitous in biology, the possibility of terrestrial contamination of meteorite samples is always a concern. Several lines of evidence can be pursued to ascertain whether amino acids are indigenous to the meteorite. Among aqueously altered carbonaceous chondrites (CR, CM, CI, CH and CB chondrites), amino acids have been found to be enriched in heavy stable isotopes including D, ^13^C, and ^15^N [37,39,40,41,42,43], whereas biology shows a strong preference for the lighter isotopes (H, ^12^C, and ^14^N) [44]. Compound-specific stable isotope measurements, therefore, are a powerful tool for establishing an extraterrestrial origin for meteoritic amino acids. There are some limitations of this technique, however. First, they require a significant amount of a given amino acid for analysis, typically >1 nmol for δ^13^C measurements, 3 nmol for δD and 6 nmol for δ^15^N, while amino acids are typically present in part-per-million (ppm) or part-per-billion (ppb) abundances [39]. This means that larger masses of irreplaceable samples must be consumed, and precludes the analysis of many low-abundance amino acids in meteorites. It has also been observed that non-α-amino acid isomers tend to be less enriched in ^13^C than their α-amino acid counterparts, consistent with different formation mechanisms from different precursor molecules [39]. This applies in particular to amino acids in thermally-altered meteorites (CV, CO, CK and other non-carbonaceous chondrite meteorites) that tend to be depleted in ^13^C [33,45]; because these amino acids are less abundant (ppb levels), and even greater amounts of sample are needed for δD and δ^15^N, often only δ^13^C measurements have been made in thermally altered meteorites. It has further been shown that thermally-driven, carbon-carbon bond-forming reactions lead to negative isotopic fractionation [46,47,48]. Stable isotope measurements may not be useful for distinguishing between terrestrial contaminants and extraterrestrial amino acids produced by thermal processes.

Other distinguishing characteristics of indigenous meteoritic amino acids are their isomeric and enantiomeric distributions as compared to amino acid distributions in biology. Aqueously altered meteorites contain broad suites of amino acids, spanning some 92 unambiguously identified amino acids that range from two to 10 or more carbons [16,49,50]. Thus far, only 12 of the 20 protein amino acids have been found in meteorites: glycine, alanine, aspartic acid, glutamic acid, serine, threonine, proline, valine, leucine, isoleucine, phenylalanine and tyrosine. The powerful separations afforded by gas chromatography and liquid chromatography-based analyses allow contamination to be assessed on a compound-specific basis. For chiral amino acids used in proteins, such as l-alanine, an assessment of its indigeneity can be made based on whether it is present in equal abundance with d-alanine (a signature of abiotic synthesis) or if the l-enantiomer is present in significant excess over the d-enantiomer (a signature of terrestrial biology). Generally, the presence of a diverse set of amino acids that includes many not commonly used in biology, with racemic ratios of chiral amino acids, is a strong indicator that the amino acids are of extraterrestrial origin, whereas meteorites that have been contaminated by biology contain predominantly l-proteinogenic amino acids [34,38]. Thermally-altered meteorites tend to contain much narrower suites of amino acids, with the predominant compounds being straight-chain amino acids including glycine, β-alanine, γ-aminobutyric acid, and δ-aminovaleric acid [33,36,45]. These compounds are achiral, so chirality cannot be used as a tool for evaluating contamination. Their relative abundances, where γ-aminobutyric acid > β-alanine > δ-aminovaleric acid > glycine are inconsistent with biological contamination as glycine is the only proteinogenic amino acid of the four. No known industrial or other processes have been determined to be a likely source of contaminants. Analyses of Antarctic ice samples from near where these meteorites were collected, and other Antarctic meteorites that contained virtually no amino acids, provide further evidence that these compounds are not widespread contaminants. Thus, it has been concluded that they are likely to be extraterrestrial in origin [33,35].

## 3. Enantiomeric Excesses in Meteoritic Amino Acids

One of the most tantalizing discoveries from the analysis of meteoritic amino acids is the presence of enantiomeric excesses that do not appear to be the result of terrestrial contamination, but rather resulted from extraterrestrial processes. Because the expectation is that abiotic chemistry should produce racemic mixtures of amino acids, significant care must be taken to rule out contamination or analytical interferences that would affect measurement of amino acid ratios. In 1997, Engel and Macko [51] reported an l-alanine excess ([L − D]/[L + D]) of ~33% in the Murchison meteorite, supported by nearly equal, extraterrestrial δ^15^N values measured for each alanine enantiomer. However, it was subsequently argued by Pizzarello and Cronin that the δ^15^N measurements could have been affected by co-eluting compounds [52]. Separately, Cronin and Pizzarello reported l-enantiomeric excesses of up to 9% in several non-proteinogenic α-methyl-α-amino acids in the CM2 chondrite Murchison: isovaline, α-methylnorvaline, and 2-amino-2,3-dimethylpentanoic acid (Figure 3) [53]. Although isotopic measurements were not made, the l-excesses were argued to be indigenous based on: (1) the performance of control experiments demonstrating the results were unlikely to be the result of analytical artifacts; (2) the relative rareness of the three amino acids on Earth; and (3) the observation that proteinogenic amino acids in the same sample were found to be racemic. Pizzarello and Cronin later reported l-enantiomeric excesses of up to 9% for α-methyl-α-amino acids in the CM2 chondrites Murchison and Murray, including 2-amino-2,3-dimethyl pentanoic acid, isovaline and α-methylnorvaline, observed previously, along with α-methylnorleucine, and α-methylvaline (Figure 3) [54]. Pizzarello and co-workers observed additional enantiomeric excesses in several more α-methyl amino acids in subsequent analyses [55,56,57].

Glavin and Dworkin built on these results by focusing on the five-carbon amino acid isovaline [58]. They analyzed the amino acid suites of several different carbonaceous chondrite groups: CI1 (Orgueil); CM2 (Murchison, Lewis Cliffs [LEW] 90500, Lonewolf Nunataks [LON] 94102, Queen Alexandria Range [QUE] 99177); CR2 (Elephant Moraine [EET] 92042). Intriguingly, isovaline l-excesses of up to 15% and 18% were found in Orgueil and Murchison, respectively, with a smaller l-excess (3.3%) in LEW 90500, whereas isovaline was found to be racemic within experimental error in LON 94102 and the two CR2 chondrites. Subsequently, Glavin and co-workers examined additional samples of aqueously altered (type 1) meteorites [15] (Figure 4). Taken together, these two studies identified a correlation between the extent of aqueous alteration a meteorite parent body experienced and the magnitude of isovaline l-enantiomeric excesses, suggesting that parent body processing might be responsible for creating or amplifying the initial enantiomeric excesses. Although the isovaline in these meteorites appeared to be extraterrestrial in origin, a further effort to confirm the extraterrestrial origin of isovaline, and rule out the possibility of terrestrial contamination was made. The most likely potential source of isovaline contamination was from fungal peptides, which contain isovaline and another amino acid commonly found in meteorites, α-aminoisobutryic acid [59]. Elsila and co-workers measured the enantiomeric composition, and δ^13^C and δ^15^N isotopic ratios of isovaline and other amino acids in several fungal peptides and in three carbonaceous chondrites [60]. They found that isovaline in the fungal peptides was present only as the d-enantiomer, and that the isotopic ratios of isovaline in the fungal peptides (−25 to −16‰ δ^13^C and +9‰ δ^15^N) were significantly lower than the corresponding values for isovaline in the meteorite samples (−5 to +51‰ δ^13^C and +68 to +77‰ δ^15^N). Thus, contamination from fungal sources was not a plausible explanation for the observed l-excesses.

Additional meteorites were analyzed for amino acids, and additional l-isovaline enantiomeric excesses were observed. In CH3 and CB chondrites, these ranged from 5–20% for the former and 10–14% for the latter [37]. Tagish Lake, an anomalous type 2 chondrite that has not been assigned to any of the existing carbonaceous chondrite groups, was found to have isovaline enantiomeric excesses ranging from 0–7%, with enantiomeric excesses appearing to correlate with increasing aqueous alteration of the subsamples that were analyzed [61,62]. Similarly, Pizzarello and co-workers reported a correlation between the extent of hydration of meteoritic material with the magnitude of isovaline enantiomeric excess, suggesting water activity plays a role in amplifying the enantiomeric excesses [63]. The observations that isovaline l-excesses have been reported by multiple laboratories, using different instruments, methods and analytical techniques and taking great care to rule out likely sources of error and potential contaminants, overwhelmingly support the conclusions that the meteoritic isovaline is extraterrestrial in origin, and that there exists some abiotic mechanism for the generation and, potentially, amplification, of isovaline enantiomeric excesses. It is reasonable to expect that a similar analytical campaign would lead to the same conclusions about the other non-protein, α-amino-α-methyl amino acids reported by Cronin and Pizzarello, given their rareness on Earth as potential contaminants.

There have also been reports of enantiomeric excesses in proteinogenic amino acids, in addition to the above discussed Engel and Macko work. In 2008, Pizzarello and co-workers reported enantiomeric excesses of 12–14% for l-isoleucine and d-*allo*-isoleucine, respectively, in the CR2 chondrite Graves Nunatak 95229; the extraterrestrial origin of these compounds was confirmed by δ^13^C measurements [56]. Isoleucine (and 2-amino-2,3-dimethylpentanoic acid, described previously) differs from isovaline and the other α-amino-α-methyl acids in that isoleucine possesses two chiral centers; one on the α-carbon and the second on the side chain (Figure 3), meaning that there are a total of four possible stereoisomers, including enantiomers (R,R/S,S; R,S/S,R) and diastereomers (e.g., R,R/R,S; S,S/S,R). While enantiomers have identical physical and chemical properties, diastereomers do not. Pizzarello and co-workers went on to report isoleucine enantiomeric excesses of up to 50% for l-isoleucine and up to 60% for d-*allo*-isoleucine, in an analysis of several CR chondrites [50]. However, isotopic measurements establishing an extraterrestrial origin for these compounds from these specific extractions were not made, and the origin of these large enantiomeric excesses has been debated [64,65].

Also in 2012, Glavin and co-workers reported on l-aspartic acid enantiomeric excesses in the Tagish Lake meteorite of up to 60% [62]. Stable isotope measurements of δ^13^C ratios for the d- and l-aspartic acid enantiomers were identical within experimental error, and showed significant enrichment in ^13^C (+24 to +29‰), consistent with an extraterrestrial origin for these compounds. Several other proteinogenic amino acids, including glutamic acid, serine and threonine, were also found to be in l-excess, but were not sufficiently abundant for δ^13^C or other isotopic measurements to be made. In the absence of compelling evidence to the contrary, these other l-amino acid excesses would be attributed to terrestrial contamination. However, the proteinogenic amino acid alanine was found to be racemic, and both alanine enantiomers and glycine had δ^13^C values that were consistent with an extraterrestrial origin. The presence of several proteinogenic amino acids with extraterrestrial isotopic signatures suggests that the other l-amino acid enantiomeric excesses could be indigenous to the meteorite, though this has not been confirmed due to the relatively large mass of sample that would be required.

## 4. Origins of Meteoritic Amino acid Enantiomeric Excesses

Meteoritic amino acid enantiomeric excesses preserve a record of an extraterrestrial mechanism for generating non-racemic mixtures of amino acids. What that mechanism is, however, remains unknown. A number of hypotheses have been advanced. For this review, we will consider possible routes to enantiomeric excesses for aspartic acid, isoleucine, and isovaline, which we consider a proxy for the other α-methyl α-amino acids reported in enantiomeric excess.

### 4.1. Amino Acid synthesis in the Meteorite Parent Body Was Likely to be Racemic

There are several plausible synthetic routes for meteoritic α-amino acids. The most widely accepted is via the Strecker pathway, which would have occurred in the meteorite parent body (Figure 5A). Evidence for this synthetic route in aqueously altered meteorites include: the discovery of suites of α-amino and their analogous α-hydroxy acids [8,66,67,68], as well as iminodicarboxylic acids that are known by-products of laboratory Strecker reactions [69]; and the detection of the necessary Strecker reactants including hydrogen cyanide [70], ammonia [41], and aldehydes and ketones [71,72] in meteorites. In regards to aspartic acid and isovaline, the relevant Strecker precursors are 3-oxopropanoic acid and butanone, respectively, both of which are achiral and thus could not have inherited any enantiomeric excess prior to accretion. As yet, there are no known asteroid-relevant mechanisms that would lead to preferential synthesis of a specific enantiomer by the Strecker pathway. Instead, aspartic acid and isovaline made by this route were likely to be racemic. Isoleucine, on the other hand, does have a chiral Strecker precursor, 2-methylbutanal. Were it to be accreted in enantiomeric excess, then it could have directed the preferential synthesis of a specific enantiomer of isoleucine [50]. However, this raises the question of how asymmetry might be induced in 2-methylbutanal, which is currently unknown.

Another plausible route to α-amino acids that could occur within meteorite parent bodies is reductive amination (Figure 5B). In this reaction, α-keto acids, which have been found in meteorites [73], react with ammonia and then are reduced to α-amino acids. However, reductive amination cannot produce α-methylamino acids such as isovaline, and the relevant aspartic acid precursor, oxaloacetic acid, is achiral. The precursor for isoleucine, 3-methyl-2-oxopentanoic acid, is chiral and, as in the Strecker scenario for isoleucine, could lead to preferential synthesis of a given enantiomer. However, the initial asymmetry in 3-methyl-2-oxopentanoic acid would somehow need to be generated for this mechanism to be viable. Another potential reaction that could occur with α-keto acids is decarboxylative transamination, where an existing amino acid would effectively transfer its amino group to the α-keto acid to make an imine that could then be reduced to an amino acid. In the presence of copper ions, it was demonstrated that this could be done in an enantioselective manner, where, beginning with nearly enantiopure l-α-methylvaline or l-α-methylleucine and the appropriate keto acid precursor, l-enantiomeric excesses of phenylalanine (37%), valine (20%) and alanine (23%) were observed in the products [74]. Unfortunately, this scenario requires rather high enantiomeric excesses for the α-methyl amino acids and yields smaller chiral excesses in the newly formed amino acids, making this more of an important demonstration for the transfer of existing enantiomeric excesses than a viable mechanism for initiating chiral asymmetry.

It has also been proposed that amino acids could be generated from aromatic compounds, such as polycyclic aromatic hydrocarbons (PAHs), during parent body aqueous alteration [75]. However, this synthetic route involves significant reprocessing of the PAHs with water, carbon dioxide, and ammonia, and it is unclear whether and how chiral selectivity could be exerted. Similarly, Fisher Tropsch-type reactions with ammonia or nitrogen, carbon monoxide or carbon dioxide, and water have been shown to lead to the production of amino acids [76,77]. Although this process are generally regarded as having occurred in the solar nebula, they could have been driven by parent body heating. However, no evidence for amino acid enantiomer excesses has been observed.

### 4.2. Relevant Mechanisms for Generating Amino Acid Enantiomeric Excesses

Several abiotic mechanisms for generating enantiomeric excesses have been identified, varying in the magnitude of enantiomeric excess they can produce, where they would occur, and whether they are biased towards a specific enantiomer (summarized in Table 2). One promising avenue for the generation of amino acid enantiomeric excesses outside the meteorite parent body involves asymmetric synthesis or destruction of amino acids in the presence of chiral light. Flores and co-workers demonstrated that enantiomeric excesses ([L − D]/[L + D]) of up to 2.5% could be generated by asymmetric photolysis of a racemic mixture of D,L-leucine with UV-circularly polarized light (UV-CPL), concomitant with the destruction of ~75% of the starting leucine [78]. Takano and co-workers were able to generate amino acid precursors through proton irradiation of gas mixtures containing carbon monoxide, ammonia and water [79]. Subsequent irradiation with UV-CPL yielded enantiomeric excesses of up to ~0.5% for alanine. Later, de Marcellus and co-workers demonstrated that alanine could be synthesized, from ices composed of water, methanol and ammonia, with small enantioenrichments of >1% during irradiation with UV-CPL (6.64 eV = 187.2 nm) [80]. It was observed that both the degree of enantioenrichment and direction of chiral excess induced (i.e., L or D) varied with the UV flux and helicity of polarization [80]. In subsequent work, Modica and co-workers demonstrated the synthesis of 16 amino acids from ice composed of water, methanol and ammonia with UV irradiation [81]. They determined the enantiomeric composition of five of those amino acids: alanine, 2,3-diaminopropanoic acid, 2-aminobutanoic acid, valine, and norvaline [81]. Intriguingly, the chirality of enantioenrichment was the same for each of the five amino acids (i.e., all had d- or all had l-excesses). However, it was found that the sign of the induced enantiomer excess reversed when irradiation was performed with polarized light of a different energy (i.e., enantiomeric excesses induced by 6.6 eV light were the opposite of the enantiomeric excesses generated by 10.2 eV [121.6 nm] light). Although isovaline was not reported as a product in these experiments, other branched chain amino acids such as α-aminoisobutyric acid, β-aminoisobutyric acid, and valine were produced. Thus, this could be a plausible mechanism for isovaline synthesis, as well. However, because isovaline was not among the reported products, it was not determined whether it is formed in enantiomeric excess during irradiation with UV-CPL. Isovaline may interact differently with UV-CPL because of the presence of the α-methyl group rather than an α-H.

If it is determined that enantiomeric excesses for isovaline can be induced by UV-CPL, it will be important to know if the direction of the induced chirality excess is the same as alanine, valine and the other three amino acids. Polarimetry measurements at 589 nm, routine analyses for determining the optical purity of chiral organic compounds, suggest this may be the case, as enantiopure (>98%) samples of the same enantiomer of alanine, valine, norvaline, 2,3-diaminopropanoic acid, and 2-aminobutanoic acid all rotate light in the same direction (i.e., the l-enantiomers of each amino acid are dextrotoratory or [+]; data obtained from commercial vendor websites). Auspiciously, l-isovaline is also dextrorotatory, suggesting that it might exhibit the same enantioenrichment behavior as the other five amino acids in a UV-CPL experiment. A complicating factor for use of 589 nm polarimetry data is that the sign, and to a lesser extent, magnitude of optical rotation are sensitive to solution conditions such as pH and acid or base concentration [82,83,84,85,86,87,88,89,90,91,92,93] (Figure 6). However, if it can be confirmed that polarimetry measurements at 589 nm are indeed predictive of UV-CPL behavior, this could be a major breakthrough for our understanding of the potential role of UV-CPL in the evolution of homochirality on Earth because polarimetry experiments can be performed routinely on standard laboratory experiments, whereas UV-CPL experiments require synchrotron beam sources. It is interesting to note that the direction of light rotation (levorotatory [−] or dextrorotatory [+]) does not necessarily correspond to amino acid chirality (L or D). For the 19 l-amino acids found in proteins, approximately half are dextrorotatory and half are levorotatory in water (Figure 6). Were optical activity at 589 nm found to be predictive of activity under UV-CPL conditions, it would be expected that the enantiomeric excesses induced in a diverse suite of amino acids would not all be the same sign, but rather would be expected to yield a mix of l- and d-enantiomeric excesses.

The astrophysical environment where UV-CPL-induced chiral excesses could plausibly affect meteoritic amino acids relevant to our Solar System are well-described elsewhere [80] and references therein. Of note for this review, however, is when enantiomeric excesses would have been generated. Because UV-CPL would not penetrate very far into an asteroid or comet, any chiral influence exerted by UV-CPL would have to have occurred prior to parent body accretion. Dust grains bearing water, ammonia and methanol or other carbon-bearing ices that were exposed to UV-CPL would provide the necessary reactants for asymmetric amino acid synthesis. After accretion, the amino acids would be subject to further processing during aqueous alteration, which could have resulted in racemization or enantiomeric excess amplification.

Another phenomenon that has been proposed for initiation of the amino acid enantiomeric excesses observed in meteorites is based on very small (picojoule per mole to femtojoule per mole) parity-violating energy differences (PVED) between enantiomers of chiral molecules, reviewed in [94]. Computational studies have shown that for several amino acids found in l-enantiomeric excess in the Murchison meteorite (isovaline, α-methylnorvaline, and 2-amino-2,3-dimethylpentanoic acid), the l-enantiomers of these compounds are slightly more thermodynamically stable than their d-counterparts [95]. Though the chiral excesses generated by PVED are expected to be small (<<1%), it was argued that they could be amplified to the levels observed in meteorites through known mechanisms. Because these are intrinsic energy differences, chiral excesses could be generated during amino acid synthesis or destruction prior to accretion or within the meteorite parent body.

An alternative mechanism for generating enantiomeric excesses relies on asymmetric destruction of d-amino acids mediated by radioactive decay products. In this scenario, ^14^N nuclei of amino acids would absorb antineutrinos emitted from a collapsing star, converting them into ^14^C nuclei, eliminating the amino group of the amino acid [96,97]. Presumably this nuclear alteration would be sufficiently destabilizing to cause the molecule to rearrange or fragment, preventing amino acid regeneration by spontaneous β-decay of the ^14^C nucleus back to ^14^N. The magnetic field associated with star collapse would serve to orient the ^14^N nuclei in a chiral manner, where one enantiomer would more favorably absorb antineutrinos, and therefore be preferentially destroyed. Calculations showed that small enantiomeric excesses (up to 0.02%) for l-alanine could be generated by this method [96], which would need to be amplified to reach the levels observed in meteorites. As with UV-CPL, any enantiomeric excess generation by anti-neutrinos would need to have occurred outside the meteorite parent body. It seems plausible that this mechanism would lead to similar enantioenrichments for the other proteinogenic amino acids found in enantiomeric excess in meteorites (aspartic acid and isoleucine), but it remains to be seen how the presence of the α-methyl groups present on isovaline and most of the other amino acids found in enantiomeric excess in meteorites would affect the viability of this mechanism.

It has also been postulated that β-particles and associated Bremsstrahlung can cause preferential destruction of an amino acid enantiomer, leading to the generation of amino acid enantiomeric excesses [98,99]. A large number of experiments have been performed to test this hypothesis, on a broad suite of amino acids, using several different radioactivity sources. The literature is extensive (see [99,100,101,102,103,104,105,106,107,108] for a representative sampling), yielding either positive reports of small enantiomeric excesses (typically <1%) or products were determined to be racemic within measurement error. Definitive conclusions for the viability of this mechanism have proven elusive because of difficulties associated with measuring small enantiomeric excesses, but the preponderance of evidence suggests that β-decay can in fact lead to amino acid asymmetry, at least to a small extent (<1%), which would need to undergo subsequent amplification to match enantiomeric excess levels observed in meteorites. Because both β-particles and associated Bremsstrahlung both drop off rapidly with distance, any chiral excesses created by this mechanism would have had to have occurred within the parent body, in close proximity to decaying radionuclides. Radioactivity sources relevant to meteorite parent bodies include ^60^Fe [109,110], which decays to ^60^Ni via ^60^Co, giving off two β-particles, an antineutrino, and Bremsstrahlung in the decay process. Nearly all of the reports in the literature have focused on proteinogenic amino acids, with only two related studies having been reported on isovaline, the only α-methyl amino acid studied [102,111]. Although it was shown that isovaline could be interconverted between enantiomers by irradiation, no net enantiomeric excesses were observed. There have been no reports to date on whether aspartic acid and isoleucine specifically can be enantioenriched by this mechanism.

### 4.3. Mechanisms to Amplify Enantiomeric Excesses

The mechanisms described above offer the possibility of breaking free from the typical racemic synthesis expected for abiotic chemistry. However, the magnitudes of enantiomeric excesses are generally very small. Thus, if any of these mechanisms were involved in the generation of the enantiomeric excesses observed for meteoritic amino acids, there must have been a mechanism to amplify them.

One tantalizing route for enantiomeric excess amplification invokes autocatalysis, where the product of a reaction then serves as a catalyst for its own production. With respect to chiral amplification, in the Frank model an enantiomer would preferentially catalyze formation of that enantiomer, and, in an ideal scenario, would inhibit production of the other enantiomer [112]. Soai and co-workers demonstrated experimentally that asymmetric autocatalysis could be achieved, though the reactions were not prebiotically plausible as they rely on alkylzinc compounds [113,114,115,116]. Were a similar demonstration to be made with amino acids in conditions relevant to meteorite parent bodies, this would be a powerful explanation for the observed amino acid enantiomeric excesses.

In a similar line of research, Plasson and co-workers (2004) calculated that amino acid enantiomeric excesses of up to 70% could have been achieved within a few centuries through polymerization and epimerization (reversal of chirality at one chiral center) reactions, beginning with a racemic amino acid pool [117]. Key components of this mechanism were that epimerization of amino acids in dipeptides must occur more easily than in free amino acids, and that heterochiral peptides (e.g., D-L) must be more prone to epimerization than homochiral peptides (e.g., L-L). Also necessary for this mechanism were appreciable rates of amino acid activation and polymerization to generate the initial peptides, and peptide hydrolysis to release enantioenriched monomers to further drive chiral amplification. Because low levels of peptides have been reported in meteorites have been reported [118], this mechanism could be plausible for meteorite parent bodies, and warrants further exploration.

Other abiotic processes for enantiomeric excess amplification involve chiral selectivity during phase transitions (solid-solution or solid-gas phase), which likely would have had to have taken place inside the meteorite parent body. Viedma and co-workers showed that amino acid chiral excesses could be amplified during sublimation [119,120], as well as in saturated solutions of amino acids [121,122]. Breslow and co-workers also showed that phenylalanine, tryptophan, and alanine enantiomeric excesses could be amplified significantly from starting enantioenrichments of just 1% by performing serial crystallization and separation of the solution phase [74,123]. These experiments exploited differences in the stability of racemic crystals versus enantiopure crystals for amino acids. Many amino acids preferentially form racemic crystals, where l- and d-enantiomers are present in a 1:1 ratio in the solid phase [124]. If an enantiomer is present in excess, any additional precipitation of that enantiomer would occur as enantiopure crystals (i.e., all L or all D). These enantiopure crystals are more prone to sublimate or dissolve in aqueous solution, and can then be transported away from the racemic solid phase in either the gas or solution phase, respectively. The end result of this process is a physical separation of an enantioenriched pool of an amino acid from a racemic pool of the same amino acid (Figure 7).

Other amino acids preferentially form enantiopure crystals, and thus cannot generate physically separated pools of enantioenriched and racemic amino acids. However, because larger crystals have lower surface area to volume ratios, larger crystals tend to dissolve more slowly than smaller ones via Ostwald ripening [125,126,127]. This phenomenon has been exploited to amplify chiral excesses in saturated solutions. Specifically, when saturated solutions of an amino acid were exposed to racemizing conditions and mechanical disruption of crystals, Noorduin and co-workers were able to achieve significant enantioenrichment of an amino acid derivative starting with a small imbalance [128,129,130]. This culminated in the demonstration that complete enantioenrichment of an amino acid derivative, beginning with a UV-CPL-generated imbalance, could be achieved. Tsogoeva and co-workers demonstrated that facile racemization could be achieved through reversible Mannich reactions, which were exploited to achieve enantiomeric excesses greater than 99% [131]. Though amino acids were not used, this provides a proof-of-concept for chiral amplification via reversible chemical reactions.

The crystallization behavior of three of the amino acids found in enantiomeric excess in meteorites, namely aspartic acid, isoleucine and isovaline, has already been determined in simple saturated solutions of water and a single amino acid. Isovaline was found to preferentially form enantiopure crystals when both enantiomers were present [132], isoleucine formed racemic crystals [124], and aspartic acid formed either enantiopure or racemic crystals depending on conditions [133]. In principle, based on their crystallization behavior it would then appear that isovaline and aspartic acid could have had their enantiomeric excesses amplified by Ostwald ripening-like processes to the levels of enantiomeric excess, whereas isoleucine could only have been physically separated into a racemic pool of amino acids (solid phase) and an enantioenriched pool via movement of the solution phase. Between isovaline and aspartic acid, an important difference is that isovaline is much more difficult to racemize than aspartic acid, and would likely need to have been mediated by harsher processes such as radioactive decay [100]. This slower racemization could help explain why isovaline enantiomeric excesses in Tagish Lake were lower than those observed for aspartic acid. An important caveat to the crystallization-based chiral amplification mechanisms, however, is the observation that the presence of additional components in solution can change the crystallization behavior of organic compounds e.g., [134]; meteorite parent bodies contain incredibly rich mixtures of organic compounds e.g., [135] and inorganic components. Nevertheless, in the absence of a robust method for generating large enantiomeric excesses of amino acids in meteorites, amplification of smaller enantiomeric excesses by crystallization is at present the strongest available hypothesis to explain the observed meteoritic amino acid enantiomeric excesses. The observation that the magnitude of enantiomeric excess, particularly for isovaline, appears to be correlated to the amount of water activity that occurred on the meteorite parent body [15,16,17,37,58,61,62,63], is consistent with this hypothesis, though more detailed experimental studies are needed.

## 5. Conclusions

Carbonaceous chondrite meteorites, which are among the most primitive Solar System materials available for laboratory study, have been found to contain a wide range of amino acids of extraterrestrial origin. For the chiral amino acids, most are racemic, as would be expected from abiotic chemistry. However, several meteorites contain one or more amino acids that are present in enantiomeric excess that do not appear to be the result of terrestrial contamination. Multiple mechanisms have been identified by which amino acid enantiomeric excesses could have been initially generated, but the magnitude of enantiomeric excesses produced by these mechanisms are generally over an order of magnitude lower than what has been observed in meteorites. Known mechanisms for amplifying amino acid enantiomeric excesses that are compatible with meteorite parent body conditions are driven by differences in crystal stability and solubility. For some amino acids, including aspartic acid and isovaline, amplification of enantiomeric excesses could be achieved through conversion of one enantiomer into the other, resulting in a net increase in the enriched enantiomer. For other amino acids, including isoleucine, enantioenrichment would be based on physical separation of enantioenriched amino acids away from a racemic solid phase, with no net increase or decrease in either enantiomer. Although the origins of amino acid enantiomeric excesses in meteorites have not yet been fully solved, the data available provide compelling evidence that abiotic chemical evolution started down the path to homochirality prior to the origins of life.

## Figures and Tables

**Figure 1 life-08-00014-f001:**
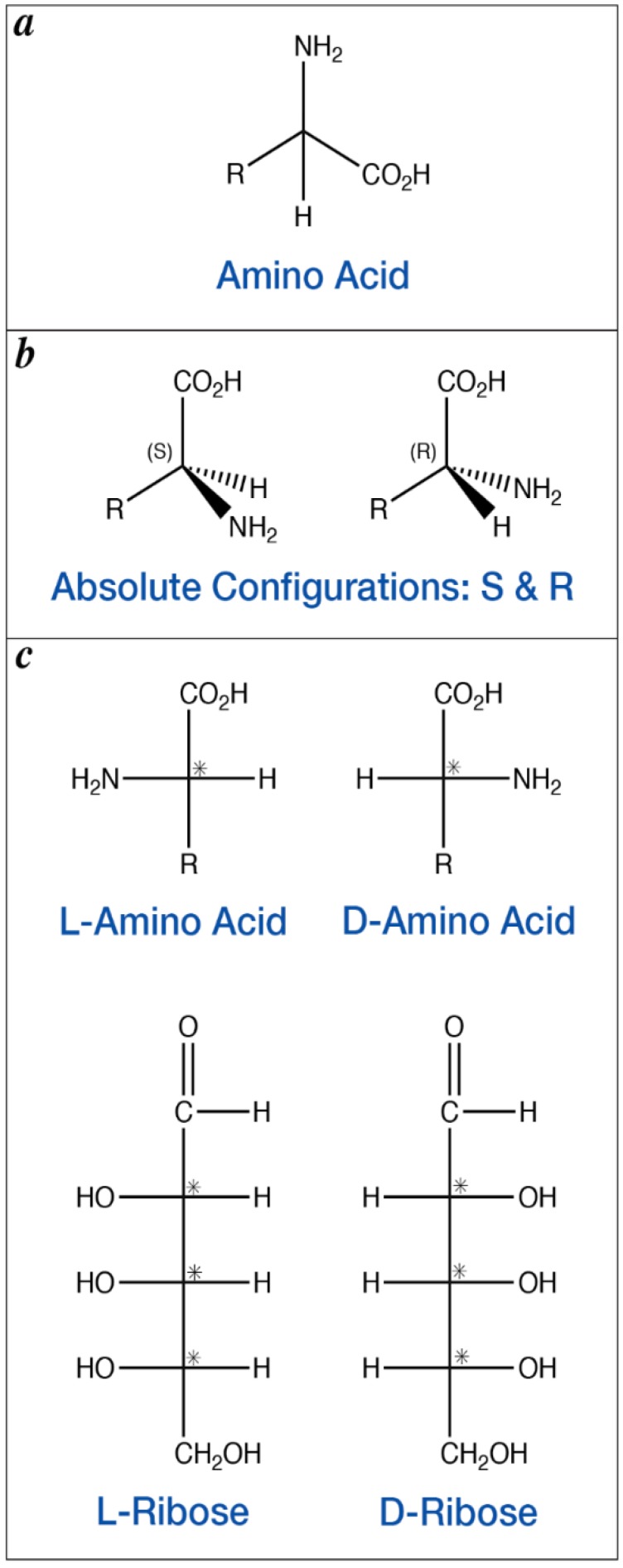
Structural drawings of amino acids and sugars. (**a**) A generic α-amino acid, where the amino group is connected to the carbon immediately adjacent to the carboxylic acid. (**b**) Stereorepresentations of the two enantiomers of a chiral α-amino acid, denoted S and R. (**c**) Fischer projections denoting chirality of a chiral α-amino acid and sugar (chiral carbons marked with asterisks), where L denotes left and D denotes right; this naming convention only reflects the first chiral center.

**Figure 2 life-08-00014-f002:**
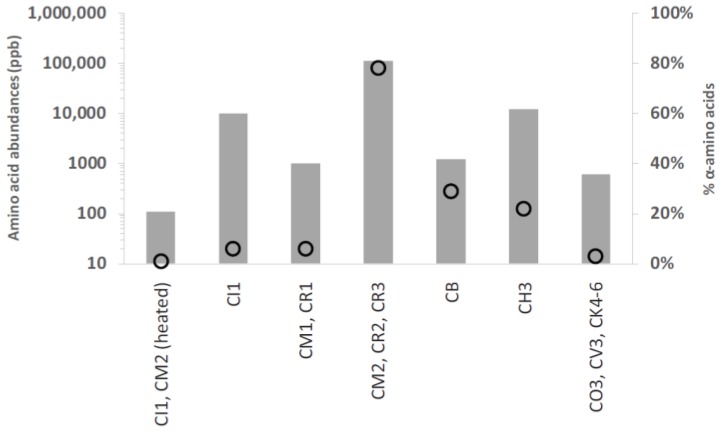
Total amino acid abundances (bars, primary axis) and percentage of five-carbon amino acids where the amino group is attached to the carbon adjacent to the carboxylic acid (α-amino acids; open circles, secondary axis) among the carbonaceous chondrite groups [15,33,34,35,36,37,38].

**Figure 3 life-08-00014-f003:**
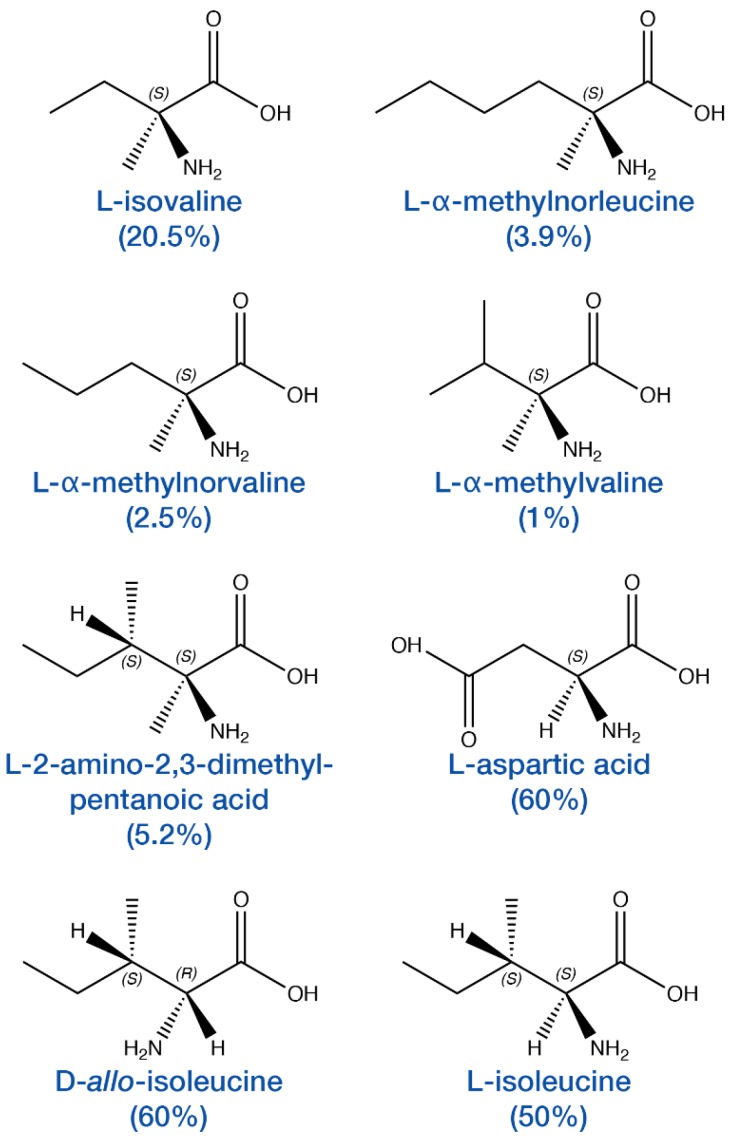
Amino acids that have been found in enantiomeric excesses >2% in multiple meteorites. Numbers in parentheses denote the highest values reported [37,50,57,62].

**Figure 4 life-08-00014-f004:**
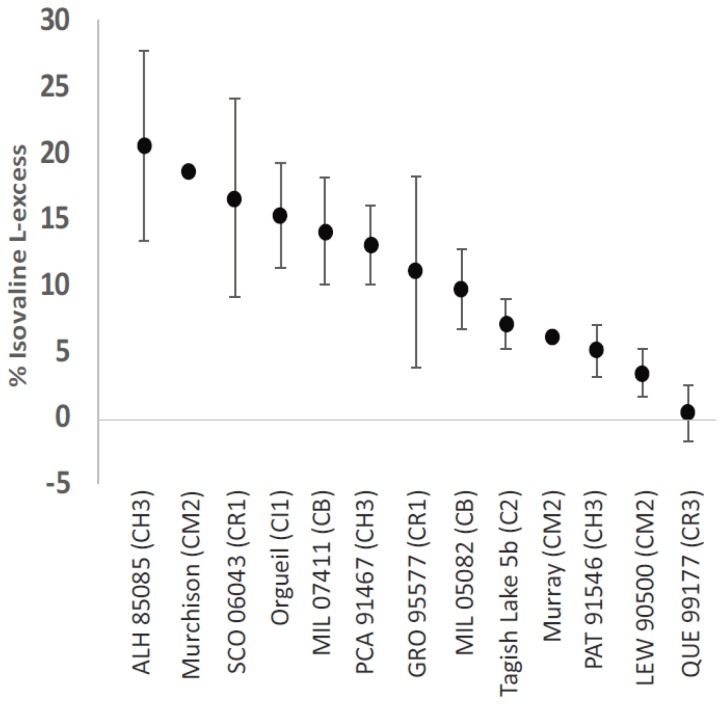
Isovaline enantiomeric excesses of 0 to 20.5% have been reported [14,37,54,61,62]. Antarctic meteorite abbreviations are Allan Hills (ALH), Scott Glaciers (SCO), Miller Range (MIL), Pecora Escarpment (PCA), Grosvenor Mountains (GRO), Patuxent Range (PAT), Lewis Cliff (LEW) and Queen Alexandria Range (QUE). The value for Murchison reflects the highest reported, but isovaline has been detected in 0 to 18% enantiomeric excess in Murchison.

**Figure 5 life-08-00014-f005:**
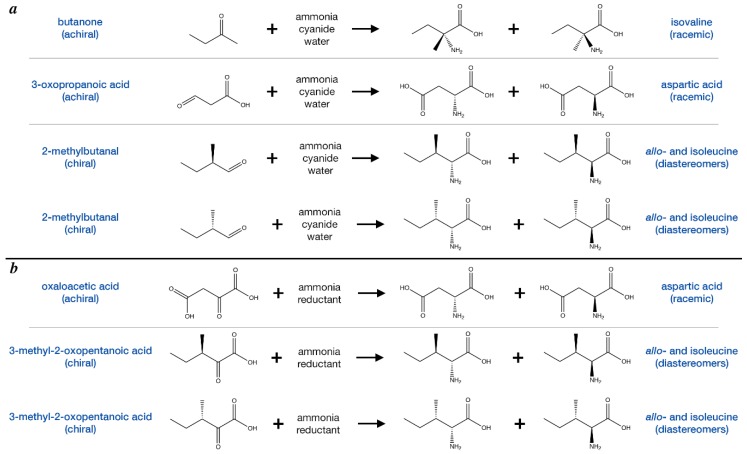
Illustration of α-amino acid synthesis routes that could have taken place in meteorite parent bodies. (**a**) The Strecker-cyanohydrin pathway relies on the reaction of aldehydes or ketones with ammonia, cyanide and water to produce amino acids. (**b**) Reductive amination reactions involve the reaction of α-keto acids with ammonia and a reductant in the meteorite parent body to produce amino acids.

**Figure 6 life-08-00014-f006:**
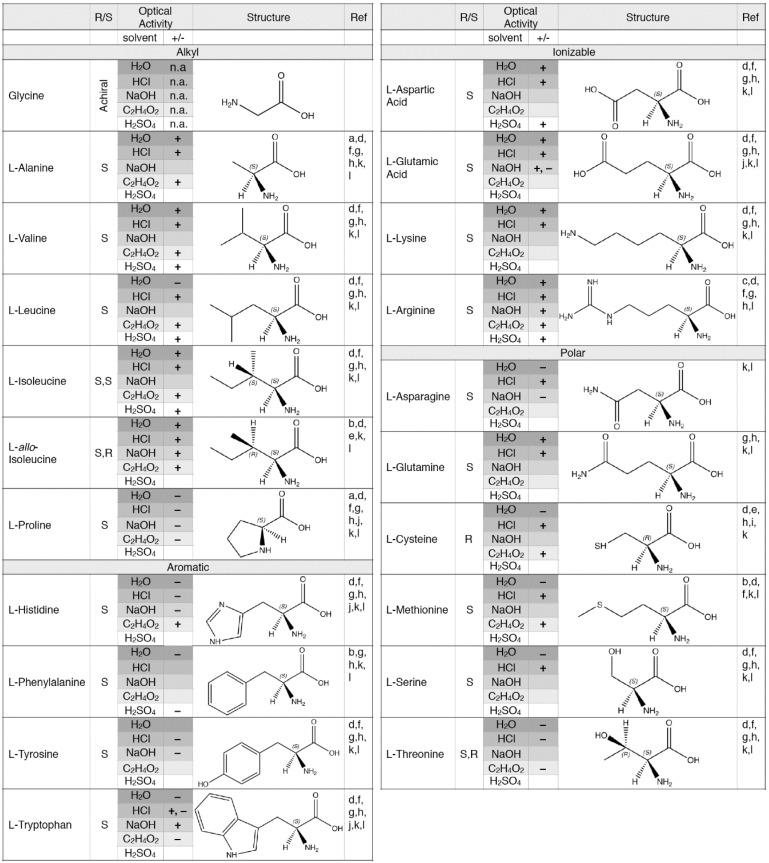
The absolute configuration (R or S) of a chiral molecule is based on the arrangement of atoms directly attached to the stereocenter. For the l-enantiomers of the amino acids listed above, all but cysteine have an absolute configuration of S. The optical activity of chiral molecules describe the direction in which each enantiomer rotates plane polarized light; clockwise is dextrorotatory (+), counterclockwise direction is levorotatory (−). The sign of the optical rotation of an enantiomer is not directly related to its absolute configuration, and can vary based on the solution in which it is measured. In addition to being solution-dependent, the optical rotation varies with the molarity of each solution. For many of the amino acids, the entire range of specific rotation values fall in the positive or negative realm; exceptions to this include l-Glutamic Acid and l-Tryptophan, which have specific rotation values that vary between negative and positive numbers in specific solvents. l-cysteine, l-isoleucine, and l-*allo*-isoleucine are dextrorotatory in glacial acetic acid. References a-k correspond to [82,83,84,85,86,87,88,89,90,91,92,93].

**Figure 7 life-08-00014-f007:**
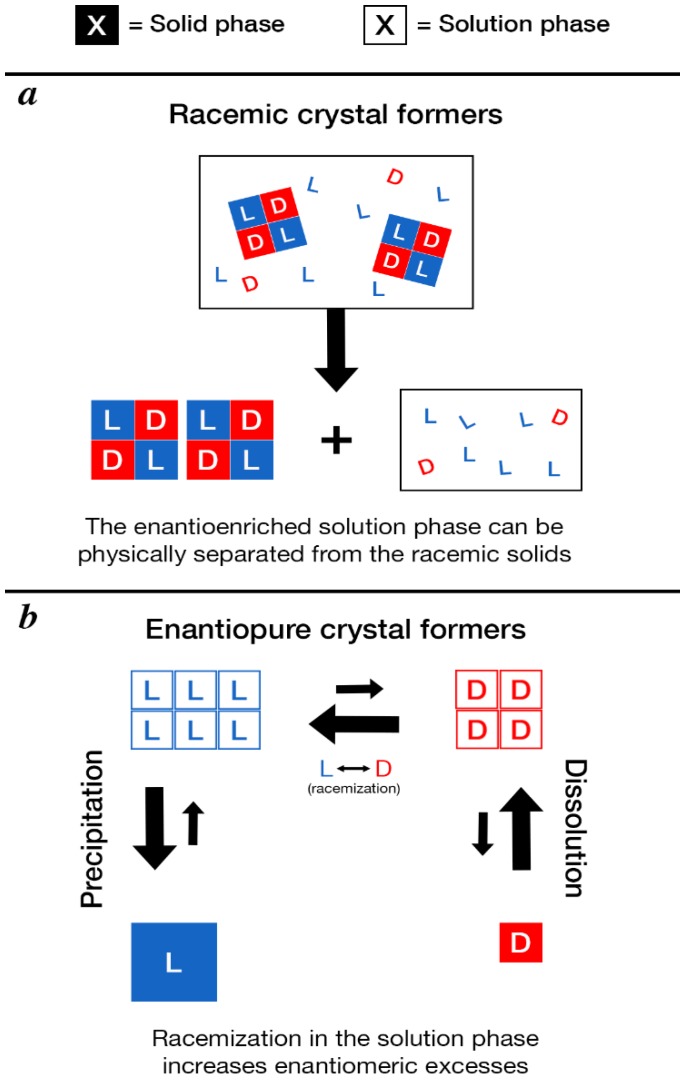
Schematic of crystallization-based enantioenrichment mechanisms. (**a**) For amino acids that preferentially form racemic crystals, in a saturated solution with one enantiomer present in excess, the solid phase would be racemic and the solution phase would be enriched to a greater degree than the overall enantiomeric excess (counting both the solution and solid phase). The enantioenriched solution phase can be physically separated from the racemic solid during aqueous alteration. (**b**) For amino acids that preferentially form enantiopure crystals, in a saturated solution where one enantiomer is present in excess, that enantiomer will tend to form more crystals that are larger, while the less abundant enantiomer will form fewer crystals that are smaller. In this case the solution phase will be racemic, so racemization would affect both enantiomers equally. However, producing more of the more abundant enantiomer will cause more of it to precipitate, increase the enantiomeric excess.

**Table 1 life-08-00014-t001:** Properties of carbonaceous chondrite meteorite groups, including: matrix abundances, chondrule abundances and sizes, refractory component abundances, metallic Fe and Ni abundances, average olivine compositions, and refractory lithophile element abundances. The carbonaceous chondrite meteorite groups are arranged from left to right (CI to CB) in order of decreasing bulk rock oxidation. Data were compiled from and the table modeled after Brearley and Jones [29], Wesiberg et al. [30], Scott and Krot [31], and Scott [32].

	CI	CM	CK	CV	CO	CR	CH	CB
Petrologic type	1	1–2	3–6	2–3	3	1–3	3	3
Chondrule abundance (vol. %)	≪1 ^†^	20 ^‡^	15	45	40–48	50–60	~70	20–40
Matrix abundance (vol. %)	>99 ^†^	70 ^‡^	75	40	30–34	30–50	5	<5
Refractory abundance ^⧺^ (vol. %)	≪1	5	4	10	13	0.5	0.1	<0.1
Metal (Fe, Ni) abundance (vol. %)	≪1	0.1	≪1	0–5	1–5	5–8	20	60–80
Avg. chondrule diameter (mm)	n.a.	0.3	0.7–0.8	1.0	0.15	0.7	0.02–0.09	0.2–10
Olivine composition								
-(mol % Fe_2_SiO_4_; range)	*	*	<1–47	*	*		<1–36	2–3
-(mol % Fe_2_SiO_4_; mode)			29–33			1–3	2	3
Refractory Lithophiles ^∦^	1.00	1.15	1.21	1.35	1.13	1.03	1.00	1.0–1.4

^†^ Including chondrule fragments and silicate minerals inferred to be chondrule fragments changes matrix and chondrule abundances to >95 vol. % and <5 vol. %, respectively; ^‡^ Variable; ^⧺^ Calcium Aluminum Inclusions + Amoeboid Olivine Aggregates; * Highly variable and unequilibrated; ^∦^ Mean ratio refractory lithophiles relative to Mg, normalized to CI chondrites.

**Table 2 life-08-00014-t002:** Summary of abiotic mechanisms for generation of amino acid enantiomeric excesses.

Mechanism	Maximum Reported Enantiomeric Excess	Chiral Preference	Influence Exerted Inside or Outside Meteorite Parent Body
Ultra-Violet circularly polarized light (UV-CPL)	2.5%	Dependent on chirality of light	outside
Parity violating energy differences (PVED)	<0.01%	L	both
Destruction of ^14^N nuclei by stellar anti-neutrinos	0.02%	L	outside
Irradiation with radioactive decay products	<1%	Dependent on chirality of radiation	inside

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
