# Peer review of "Insights into Abiotically-Generated Amino Acid Enantiomeric Excesses Found in Meteorites"

_life, 2018, doi:10.3390/life8020014_

Round 1
Reviewer 1 Report
The review article "Insights into Abiotically-Generated Amino Acid Enantiomeric Excesses Found in Meteorites" is interesting and well written. There is only one minor revision to be made:
I think that in line 348 page 11, "be" is missing.
Author Response
We did indeed omit the necessary "be" in line 348 and have added it.
Reviewer 2 Report
The manuscript submitted by Dr. Burton and Dr. Berger constitutes an attempt to gather information on the occurrence of non-racemic mixture of amino acids in carbonaceous chondrites. The manuscript is clearly written. Although, this is not my field of expertise I read with interest the sections on meteorite classification and the analytical problems. I think this manuscript constitutes a nice overview of the work carried out on this topic that should be useful for Life’s general audience and is worth to be published provided that some minor issues are addressed by the authors. The comments are listed below from the more important to the less important:
1) Section on the mechanism for amplifying enantiomeric excesses (line 435 and following): the authors do not mention the possibility that a non-racemic steady state could result from peptide formation as for instance in the model proposed by Plasson et al. (PNAS 2004), which should be considered since the presence of peptides in peptides has been often been suggested. More generally, the formation of polymers or at least covalent adducts is a usual way to reveal differences between enantiomers that could result in symmetry breaking or the amplification of imbalances.
2) Section 4.1 (line 285). Strecker synthesis should not be expressed as the “Strecker cyanohydrin pathway”, which is misleading since cyanohydrin is a side-product rather than an intermediate in the process.
3) Line 435. The section is erroneously numbered 4.2 instead of 4.3.
Author Response
1) Section on the mechanism for amplifying enantiomeric excesses (line 435 and following): the authors do not mention the possibility that a non-racemic steady state could result from peptide formation as for instance in the model proposed by Plasson et al. (PNAS 2004), which should be considered since the presence of peptides in peptides has been often been suggested. More generally, the formation of polymers or at least covalent adducts is a usual way to reveal differences between enantiomers that could result in symmetry breaking or the amplification of imbalances.
This is an excellent point by the reviewer. We have added the following paragraph below the autocatalysis discussion on page 15:
In a similar line of research, Plasson and co-workers (2004) calculated that amino acid enantiomeric excesses of up to 70% could have been achieved within a few centuries through polymerization and epimerization (reversal of chirality at one chiral center) reactions, beginning with a racemic amino acid pool [117]. Key components of this mechanism were that epimerization of amino acids in dipeptides must occur more easily than in free amino acids, and that heterochiral peptides (e.g., D-L) must be more prone to epimerization than homochiral peptides (e.g., L-L). Also necessary for this mechanism were appreciable rates of amino acid activation and polymerization to generate the initial peptides, and peptide hydrolysis to release enantioenriched monomers to further drive chiral amplification. Because low levels of peptides have been reported in meteorites have been reported [118], this mechanism could be plausible for meteorite parent bodies, and warrants further exploration.
2) Section 4.1 (line 285). Strecker synthesis should not be expressed as the “Strecker cyanohydrin pathway”, which is misleading since cyanohydrin is a side-product rather than an intermediate in the process.
We have accepted the reviewer's suggestion and removed cyanohydrin from the text.
3) Line 435. The section is erroneously numbered 4.2 instead of 4.3.
We have corrected the error and thank the reviewer.
Reviewer 3 Report
I like this review. It is well-written and easy to understand. I'm happy to recommend it for publication, almost as is. My only recommendations are two small expansions:
Lipids are also chiral (along with the amino acids and sugars). These might be worth discussing in section 1 as they add to the number of biomolecules with chiral arrangement. However, lipids also serve as a demonstrative case where chirality is not defined by LUCA. Both enantiomers are present in the domains of life, though they are still deep-rooted. This shows that not all chirality had to be selected for early on, and some came after the development of life (Peretó, J., López-García, P., & Moreira, D. (2004). Ancestral lipid biosynthesis and early membrane evolution. Trends in biochemical sciences, 29(9), 469-477.)
A brief discussion of the locations where amino acids have been proposed to form should also include aqueous alteration on the parent body (e.g., Shock and Schulte 1990). Intriguingly, your article can be used to dismiss this route to some extent, given the requirements of the penetration of radiation for formation of enantiomeric excesses.
Beyond those minor comments/additions, this paper provides a good review of amino acids and their chirality, and will be useful for the students of the field.
Author Response
#1 Lipids are also chiral (along with the amino acids and sugars). These might be worth discussing in section 1 as they add to the number of biomolecules with chiral arrangement. However, lipids also serve as a demonstrative case where chirality is not defined by LUCA. Both enantiomers are present in the domains of life, though they are still deep-rooted. This shows that not all chirality had to be selected for early on, and some came after the development of life (Peretó, J., López-García, P., & Moreira, D. (2004). Ancestral lipid biosynthesis and early membrane evolution. Trends in biochemical sciences, 29(9), 469-477.)
This is a very interesting point. We have updated the text on page 2 to the following:
In contrast with amino acids and sugars, both enantiomers of chiral phospholipids (a major component of cell membranes), are used in extant biology, with archaea and bacteria using opposite enantiomers [4]. In this case, selection occurred post-LUCA. While contemporary biology tells us which stereochemistries were ultimately selected for, and provides a constraint on when that selection occurred (prior to LUCA, >3.5 Ga for amino acids and sugars, and post-LUCA for phospholipids), analysis of modern biology reveals little about how and why L-amino acids and D-sugars were ultimately selected over their respective enantiomers, necessitating an alternative approach.
#2 A brief discussion of the locations where amino acids have been proposed to form should also include aqueous alteration on the parent body (e.g., Shock and Schulte 1990). Intriguingly, your article can be used to dismiss this route to some extent, given the requirements of the penetration of radiation for formation of enantiomeric excesses.
We have taken the reviewer's suggestion and incorporated a brief discussion of the Shock and Schulte work along with Fischer Tropsch-type chemistry leading to amino acids on page 10:
It has also been proposed that amino acids could be generated from aromatic compounds, such as polycyclic aromatic hydrocarbons (PAHs), during parent body aqueous alteration [75]. However, this synthetic route involves significant reprocessing of the PAHs with water, carbon dioxide, and ammonia, and it is unclear whether and how chiral selectivity could be exerted. Similarly, Fisher Tropsch-type reactions with ammonia or nitrogen, carbon monoxide or carbon dioxide, and water have been shown to lead to the production of amino acids [76,77]. Although this process are generally regarded as having occurred in the solar nebula, they could have been driven by parent body heating. However, no evidence for amino acid enantiomer excesses has been observed.